**Data Availability Statement:** The data that support the findings of this study are available in GitHub repository at https://github.com/

# COVID-19 outcomes in hospitalized puerperal, pregnant, and neither pregnant nor puerperal women

**Fabiano Elisei Serra**[1], **Rossana Pulcineli Vieira Francisco**[1]*, **Patricia de Rossi**[2,3], **Maria de Lourdes Brizot**[1], **Agatha Sacramento Rodrigues**[1,4]*

1 Disciplina de Obstetrícia, Departamento de Obstetrícia e Ginecologia, Faculdade de Medicina da Universidade de São Paulo, São Paulo, São Paulo, Brazil, 2 Curso de Medicina, Universidade de Santo Amaro (UNISA), São Paulo, São Paulo, Brazil, 3 Gerência de Medicina Perinatal e Ginecologia, Conjunto Hospitalar do Mandaqui, São Paulo, São Paulo, Brazil, 4 Departamento de Estatística, Universidade Federal do Espírito Santo, Vitória, Espírito Santo, Brazil

* agatha.rodrigues@ufes.br (ASR); rossana.francisco@hc.fm.usp.br (RPVF)

## Abstract

### Objective

To compare hospitalized reproductive age women with COVID-19 who were pregnant, puerperal, or neither one nor the other in terms of demographic and clinical characteristics and disease progression using Brazilian epidemiological data.

### Methods

A retrospective analysis of the records of the Information System of the Epidemiological Surveillance of Influenza of the Health Ministry of Brazil was performed. It included the data of female patients aged 10 to 49 years hospitalized because of severe COVID-19 disease (RT-PCR+ for SARS-CoV-2), from February 17, 2020 to January 02, 2021. They were separated into 3 groups: pregnant, puerperal, and neither pregnant nor puerperal. General comparisons and then adjustments for confounding variables (propensity score matching [PSM]) were made, using demographic and clinical characteristics, disease progression (admission to the intensive care unit [ICU] and invasive or noninvasive ventilatory support), and outcome (cure or death). Deaths were analyzed in each group according to comorbidities, invasive or noninvasive ventilatory support, and admission to the ICU.

### Results

As many as 40,640 reproductive age women hospitalized for COVID-19 were identified: 3,372 were pregnant, 794 were puerperal, and 36,474 were neither pregnant nor puerperal. Groups were significantly different in terms of demographic data and comorbidities (p<0.0001). Pregnant and puerperal women were less likely to be symptomatic than the women who were neither one nor the other (72.1%, 69.7% and 88.8%, respectively). Pregnant women, however, had a higher frequency of anosmia, and ageusia than the others. After PSM, puerperal women had a worse prognosis than pregnant women with respect to

observatorioobstetrico/paper_covid19_groups.
These data were derived from the following
resources available in the public domain: https://
opendatasus.saude.gov.br/dataset/bd-srag-2020
obtained on January 11, 2021.

**Funding:** This work was supported, in whole or in
part, by the Bill & Melinda Gates Foundation [INV-
027961]. Under the grant conditions of the
Foundation, a Creative Commons Attribution 4.0
Generic License has already been assigned to the
Author Accepted Manuscript version that might
arise from this submission. This work is also
funded by CNPq (Award Number: 445881/2020-8)
and FAPES (Award Number: 007/2021). The
funders had no role in study design, data collection
and analysis, decision to publish, or preparation of
the manuscript.

**Competing interests:** The authors have declared
that no competing interests exist.

admission to the ICU, invasive ventilatory support, and death, with OR (95% CI) 1.97 (1.55 − 2.50), 2.71 (1.78 − 4.13), and 2.51 (1.79 − 3.52), respectively.

## Conclusion

Puerperal women were at a higher risk for serious outcomes (need for the ICU, need for invasive and noninvasive ventilatory support, and death) than pregnant women.

## Introduction

COVID-19 is an infectious disease caused by the new coronavirus (SARS-CoV-2), and it has a clinical spectrum ranging from absence of symptoms to severe illness and death. Widespread contagion and the ability of the virus to disseminate led the World Health Organization (WHO) to declare a pandemic state in March 2020 [1]. The high incidence has had a tremendous socioeconomic impact worldwide. One day prior to the submission of this article (June 28, 2021), globally, there were more than 180 million confirmed cases of COVID-19 and 3,923,238 deaths [2].

Since the beginning of the pandemic, the infection has been studied in the obstetric population to understand its consequences and to prevent adverse maternal-fetal outcomes. The initial publications describing COVID-19 cases in pregnant women in China, Europe, and North America did not report increased severity and deaths compared to the general population [3–9]. Subsequent studies, however, showed a higher likelihood of the need for admission to intensive care units (ICUs) and mechanical ventilation [10–14]. The first maternal deaths were reported in the United Kingdom, Iran, United States, Mexico, and France [15–19]. In Brazil, the high number of maternal deaths due to COVID-19 has been attributed to factors such as high birth rate, poor nutrition and health, difficult access to health services, and insufficient obstetric assistance [20, 21].

Socioeconomic heterogeneity in Brazil is reflected in the quality of health services and in the availability of hospital and ICU beds, having a great impact on health indicators of both pregnant and puerperal women [22, 23]. Understanding the disease and evaluating why the prognoses of pregnant and puerperal women have been worse in this pandemic is relevant. We did not find any studies comparing demographic and clinical characteristics and disease progression among pregnant women, puerperal women, and neither pregnant nor puerperal women hospitalized with COVID-19. Therefore, the authors of the present study aim to compare pregnant women, puerperal women, and neither pregnant nor puerperal women according to data related to the SARS-CoV-2 infection by using population statistics from SIVEP-Gripe (System of Information about Epidemiological Surveillance of Influenza) of the Health Ministry of Brazil.

## Materials and methods

A retrospective analysis of the subjects from SIVEP-Gripe, a Brazilian national database containing surveillance data on severe acute respiratory syndrome (SARS) was performed [24]. The notification of SARS is compulsory in cases of the flu syndrome (acute respiratory condition, characterized by at least two of the following signs and symptoms: fever [even if reported], chills, sore throat, headache, cough, coryza, and disorders of taste or smell), accompanied by dyspnea/respiratory distress, persistent chest pressure, oxygen saturation ($SpO_2$) below 95% in room air, or cyanosis. The SIVEP-Gripe is notified of all cases of hospitalization

both in public and in private hospitals, as well as of all deaths caused by SARS-CoV-2, irrespective of hospitalization.

SIVEP-Gripe records include the following: demographic data (sex, age, skin color/ethnicity, obstetric status, schooling, city of residence); clinical data (signs and symptoms, risk factors/comorbidities); epidemiological data (previous flu vaccination, community-acquired infection, or nosocomial infection); laboratory and etiological diagnoses. There is also information about hospital admission, ICU admission, use of ventilatory support (invasive and noninvasive), and disease outcome (cure or death).

Data search covered epidemiological weeks 1 to 53 (December 29, 2019 - January 02, 2021), with the last update on January 11, 2021; however, the first Brazilian records began in epidemiological week 8 (first day of symptoms of the first confirmed case was on February 17, 2020). Search included all data on female patients aged 10 to 49 years hospitalized with COVID-19, confirmed with a positive RT-PCR result for SARS-CoV-2. Cases were excluded if they were unhospitalized or unconfirmed with an RT-PCR for SARS-CoV-2, or if gender or pregnancy status were not recorded. The result was 40,640 women hospitalized with COVID-19 (RT-PCR+) and aged between 10 and 49 years, who were divided into two groups: pregnant women (n = 3,372) and nonpregnant women (n = 37,268). The latter were separated into puerperal (n = 794) and neither pregnant nor puerperal (n = 36,474) (Fig 1). Only valid responses of each analyzed variable are considered. The number of valid observations of each variable is always identified in the tables of analysis. Variables used in the analysis were age, pregnancy status, comorbidities, schooling, skin color/ethnicity, signs and symptoms, SARI (Severe Acute Respiratory Infection, defined as temperature $\geq 38°C$, cough, and onset in 10 days), SARI without fever, admission to ICU, respiratory support, and outcome (cure or death). Comorbidities reported were chronic cardiovascular, renal, neurological, hematologic, hepatic, and respiratory diseases, asthma, obesity, diabetes, and immunosuppression. Fever, cough, sore throat, dyspnea, respiratory discomfort, $SpO_2 < 95\%$, diarrhea, vomiting, abdominal pain, fatigue, anosmia, and ageusia were the signs and symptoms.

Deaths were analyzed individually according to comorbidities, invasive and noninvasive respiratory support, and admission to ICU.

SIVEP-Gripe records are publicly available anonymized data. Therefore, according to Brazilian Ethics regulatory requirements, there is no need for ethical approval by an Institutional Review Board.

## Data analysis

Quantitative variables were summarized as mean and standard deviation. Qualitative variables were displayed as absolute frequencies (n) and category percentages (%).

Nonparametric Kruskal-Wallis test was applied to compare the three study groups in terms of quantitative variables and pairwise comparisons using Wilcoxon rank sum test with continuity correction were considered. Chi-square test was used to evaluate the association between groups and qualitative variables. Odds Ratio (OR) was considered as a measure of association to compare the relative odds of the occurrence of the outcome of interest. Alpha adjustment for multiple comparisons through the Bonferroni method is considered. As the significance level adopted is 5% (alpha=0.05), the adjusted alpha is 0.05/3=0.016.

Propensity score matching (PSM) was used for estimating and assessing balancing weights for the observations to make the three balanced groups in relation to the confounding variables through Inverse Probability of Treatment Weighting Method (IPTW) [25]. Multinomial regression PS is the method used to create the propensity score weights and the Average Treatment Effect (ATE) is estimated for treatment effects based on IPTW. After estimating the

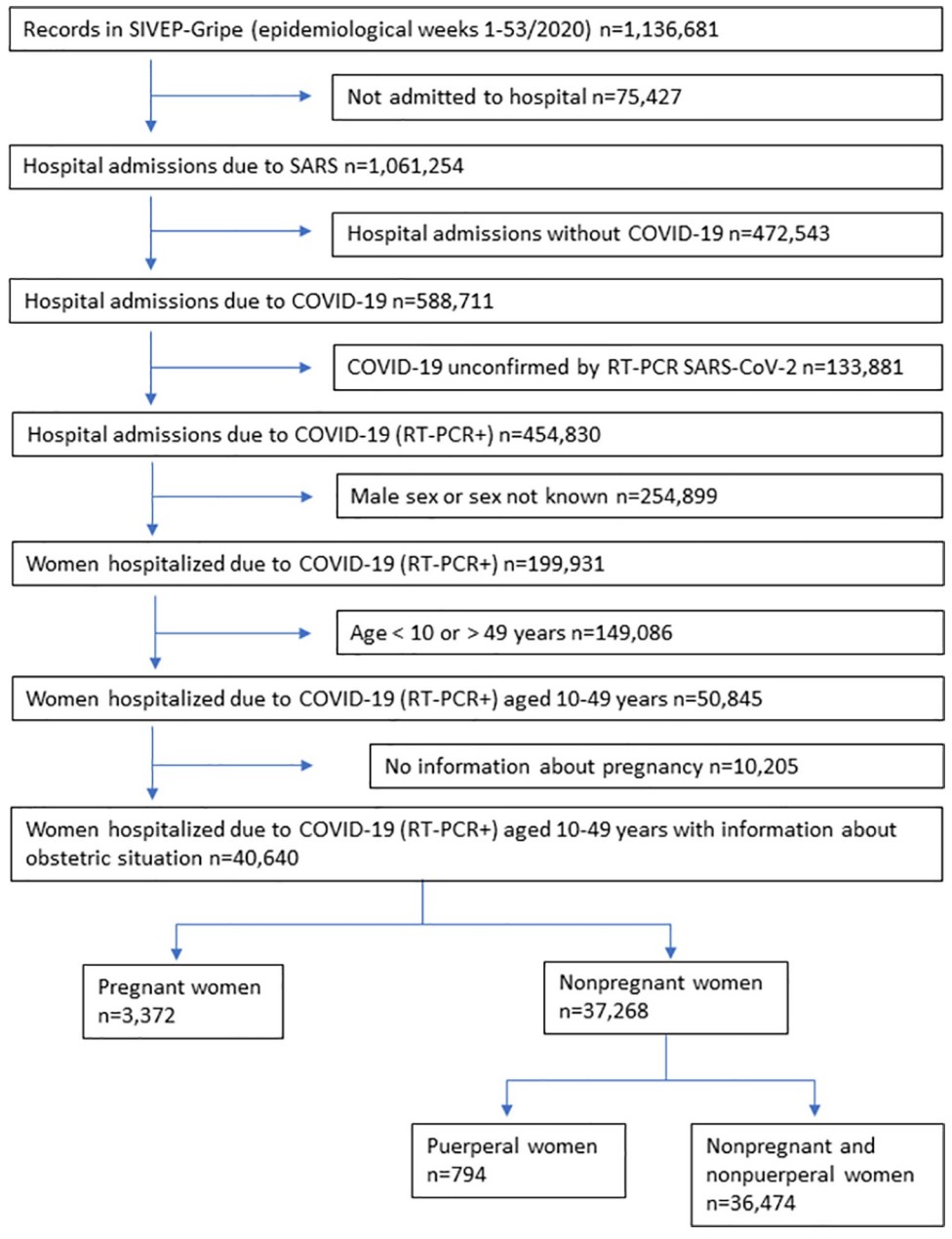

**Fig 1. Study profile.**

weights, weighted logistic regression is considered for binary variables, and the weighted multinomial log-linear model is fitted for multiple categories variables (as ventilator support). The first analysis, used to compare the symptoms between groups, included age, ethnicity, chronic cardiovascular disease, asthma, diabetes, immunosuppression, and obesity as control variables. The second one, comparing the outcomes, included the same variables plus schooling, Federative Unit of Brazil, and respiratory symptoms. The groups are balanced with respect to the control variables after considering the PSM weight. The results of PSM and standardized difference after the matching can be seen in the supplemental material.

The analyses were performed with the statistical R software (R Foundation for Statistical Computing Platform, version 4.0.3) [26] and PSM was carried out with the R Weightlt package [27].

## Results

A total of 40,640 reproductive age women hospitalized with a positive RT-PCR for SARS-CoV-2 result were identified. Of these, 3,372 were pregnant and 37,268 were not pregnant (Fig 1). Nonpregnant women comprised 794 puerperal women and 36,474 women who were neither pregnant nor puerperal. Groups differed significantly related to demographic data and comorbidities (Table 1). All three groups are different for age (neither pregnant nor puerperal vs pregnant: p-value <0.0001, neither pregnant nor puerperal vs puerperal: p-value <0.0001 and pregnant versus puerperal: p-value=0.034). The group of neither pregnant nor puerperal women had the highest rate of comorbidities.

Table 2 shows the results of the comparison of COVID-19 symptoms before and after PSM in the three study groups. Both pregnant and puerperal women had a smaller chance of fever, cough, sore throat, dyspnea, respiratory discomfort, $SpO_2$<95%, diarrhea, vomiting, fatigue, SARI, and SARI without fever than neither pregnant nor puerperal women. Pregnant women had anosmia and ageusia more frequently than the others. Puerperal women had less chance of fever, coughing, vomiting, SARI, and SARI without fever than pregnant women; however, their oxygen saturation level was more frequently lower than 95% at hospital admission.

A remarkable fact related to hospital-acquired COVID-19 is that pregnant women developed the disease less frequently (1.4%) than neither pregnant nor puerperal women (2.8%) and puerperal women (5.7%).

A PSM analysis of the statistical differences between the groups, especially regarding age, skin color/ethnicity, and comorbidities, showed that pregnant women had cough more frequently than neither pregnant nor puerperal women. Sore throat and fatigue did not differ significantly between the groups. The remainder of the PSM analysis comparing pregnant and puerperal women with neither pregnant nor puerperal yielded similar results to the general analysis.

Pregnant, compared to neither pregnant nor puerperal women, were less likely to have any of the study outcomes (admission to ICU, invasive and noninvasive respiratory support, and death). On the other hand, puerperal women, when contrasted with neither pregnant nor puerperal, were more likely to be admitted to the ICU and less likely to need noninvasive support. Puerperal had a higher risk of needing admission to the ICU, requiring invasive respiratory support, and dying, than pregnant women (Table 3).

After PSM (with the inclusion schooling, Federative Unit of Brazil, and respiratory symptoms variables), the only difference between this analysis and the general analysis was that puerperal women were no longer more likely than neither pregnant nor puerperal women to be admitted to ICU, while maintaining a worse prognosis than that of pregnant women with respect to admission to the ICU, invasive ventilatory support, and death.

Pregnant women with chronic cardiovascular or renal disease, asthma, or diabetes, and those in the ICU, or those who received ventilatory support were less likely to die than neither pregnant nor puerperal women with the same characteristics (Table 4). Puerperal women did not differ significantly from neither pregnant nor puerperal regarding the study variables. Puerperal women had a greater likelihood of dying than pregnant women when both had diabetes, or when received noninvasive ventilatory support, or when they were admitted to the ICU. After PSM, of women with chronic cardiovascular disease or diabetes and of those who received noninvasive support, pregnant were less likely to die than neither pregnant nor puerperal women.

**Table 1. Demographic and clinical characteristics of reproductive age women with COVID-19 according to gestational status – Brazil, February 17, 2020 – January 02, 2021.**

| Characteristics | Neither pregnant nor puerperal | Pregnant | Puerperal | P |
|---|---|---|---|---|
| **Age (years)** | | | | |
| **Mean ± SD** | 38.22 ± 8.25 | 29.64 ± 6.93 | 30.24 ± 7.34 | <0.0001[a] |
| **Age bracket (years)** | **n (%)** | **n (%)** | **n (%)** | |
| <20 | 1,009 (2.8) | 250 (7.4) | 61 (7.7) | <0.0001[b] |
| 20-34 | 9,629 (26.4) | 2,244 (66.5) | 494 (62.2) | |
| >34 | 2,5836 (70.8) | 878 (26.0) | 239 (30.1) | |
| Total | 36,474 | 3,372 | 794 | |
| **Skin color/Ethnicity** | **n (%)** | **n (%)** | **n (%)** | |
| White | 15,418 (54.0) | 1,102 (40.5) | 238 (37.2) | <0.0001[b] |
| Black | 1,679 (5.9) | 192 (7.1) | 46 (7.2) | |
| Yellow | 359 (1.3) | 33 (1.2) | 4 (0.6) | |
| Brown | 10,976 (38.5) | 1,382 (50.8) | 348 (54.4) | |
| Indigenous | 97 (0.3) | 13 (0.5) | 4 (0.6) | |
| Total | 28,529 (100) | 2,722 (100) | 640 (100) | |
| **Schooling** | **n (%)** | **n (%)** | **n (%)** | |
| No schooling | 219 (1.4) | 7 (0.5) | 2 (0.6) | |
| Up to high school | 3,661 (23.7) | 360 (25.1) | 81 (24.5) | <0.0001[b] |
| High school | 7,532 (48.7) | 790 (55.1) | 179 (54.2) | |
| College | 4,058 (26.2) | 277 (19.3) | 68 (20.6) | |
| Total | 15,470 (100) | 1,434 (100) | 330 (100) | |
| **Risk factors and comorbidities** | **n (%)** | **n (%)** | **n (%)** | |
| Chronic cardiovascular disease (n = 14,976) | 5780/13,330 (43.4) | 214/1,141 (18.8) | 77/505 (15.2) | <0.0001[b] |
| Chronic hematologic disease (n = 12,901) | 356/11,340 (3.1) | 16/1,081 (1.5) | 11/480 (2.3) | 0.006[b] |
| Chronic hepatic disease (n = 12,769) | 189/11,226 (1.7) | 8/1,066 (0.8) | 4/477 (0.8) | 0.0273[b] |
| Asthma (n = 13,426) | 1883/11,821 (15.9) | 139/1,119 (12.4) | 33/486 (6.8) | <0.0001[b] |
| Diabetes (n = 14,575) | 4825/12,934 (37.3) | 241/1,151 (20.9) | 61/490 (12.4) | <0.0001[b] |
| Chronic neurological disease (n = 12,933) | 568/11,377 (5.0) | 25/1,078 (2.3) | 5/478 (1.0) | <0.0001[b] |
| Chronic lung disease (n = 12,945) | 579/11,390 (5.1) | 23/1,075 (2.1) | 5/480 (1.0) | <0.0001[b] |
| Immunodepression (n = 13,137) | 1,289/11,578 (11.1) | 40/1,078 (3.7) | 18/481 (3.7) | <0.0001[b] |
| Chronic renal disease (n = 13,047) | 1,080/11,495 (9.4) | 24/1,073 (2.2) | 12/479 (2.5) | <0.0001[b] |
| Obesity (n = 13,713) | 3,741/12,135 (30.8) | 143/1,097 (13.0) | 53/481 (11.0) | <0.0001[b] |
| Metabolic syndrome (n = 12,509) | 422/10,998 (3.8) | 7/1,044 (0.7) | 7/467 (1.5) | <0.0001[b] |
| **Number of comorbidities (n = 11,958)** | **n (%)** | **n (%)** | **n (%)** | **< 0.0001[b]** |
| 0 | 2,022/10,502 (19.3) | 576/999 (57.7) | 335/457 (73.3) | |
| 1-2 | 7,681/10,502 (73.1) | 408/999 (40.8) | 106/457 (23.2) | |
| >2 | 799/10,502 (7.6) | 15/999 (1.5) | 16/457 (3.5) | |

[a] Kruskal-Wallis test

[b] Chi-square test.

## Discussion

The results of the current analysis show that at hospital admission, both pregnant and puerperal women, in general, presented a lower rate of symptoms than neither pregnant nor puerperal women, except for cough, anosmia, and ageusia of pregnant women. On the other hand, the contrast between puerperal and pregnant women revealed that the former had a higher

**Table 2. COVID-19 signs and symptoms in pregnant, puerperal, and neither pregnant nor puerperal reproductive age women before and after propensity score matching – Brazil, February 17, 2020 – January 02, 2021.**

| Signs and Symptoms | Neither pregnant nor puerperal | Pregnant | Puerperal | Group comparison | | | Group comparison after PSM (age + race + cardiopathy + asthma + diabetes + immunosuppression + obesity) | | |
|---|---|---|---|---|---|---|---|---|---|
| | n (%) | n (%) | n (%) | Pregnant vs. NPrNPu | Puerperal vs. NPrNPu | Puerperal vs. Pregnant | Pregnant vs. NPrNPu | Puerperal vs. NPrNPu | Puerperal vs. Pregnant |
| | | | | OR (98,33% CI) | OR (98,33% CI) | OR (98,33% CI) | OR (98,33% CI) | OR (98,33% CI) | OR (98,33% CI) |
| Fever (n = 36,371) | 24,297/32,702 (74.3) | 2,038/2,983 (68.3) | 424/686 (61.8) | 0.75 (0.68 – 0.82) | 0.56 (0.46 – 0.68) | 0.75 (0.61 – 0.93) | 0.78 (0.67 – 0.90) | 0.63 (0.51 – 0.77) | 0.81 (0.65 – 1.00) |
| Cough (n = 37,101) | 27,305/33,324 (81.9) | 2,408/3,085 (78.1) | 475/692 (68.6) | 0.78 (0.70 – 0.88) | 0.48 (0.40 – 0.59) | 0.62 (0.49 – 0.77) | 1.19 (1.01 – 1.41) | 0.81 (0.64 – 1.01) | 0.68 (0.54 – 0.85) |
| Sore throat (n = 31,474) | 8,854/28,242 (31.4) | 724/2,629 (27.5) | 156/603 (25.9) | 0.83 (0.75 – 0.93) | 0.76 (0.61 – 0.95) | 0.92 (0.72 – 1.17) | 0.88 (0.75 – 1.03) | 0.90 (0.71 – 1.15) | 1.02 (0.79 – 1.32) |
| Dyspnea (n = 36,272) | 25,134/32,664 (76.9) | 1,761/2,940 (59.9) | 381/668 (57.0) | 0.45 (0.41 – 0.49) | 0.40 (0.33 – 0.48) | 0.89 (0.72 – 1.09) | 0.54 (0.46 – 0.63) | 0.54 (0.44 – 0.67) | 1.00 (0.81 – 1.24) |
| Respiratory discomfort (n = 34,023) | 20,158/30,600 (65.9) | 1,374/2,771 (49.6) | 341/652 (52.3) | 0.51 (0.46 – 0.56) | 0.57 (0.47 – 0.69) | 1.11 (0.90 – 1.37) | 0.60 (0.52 – 0.69) | 0.73 (0.59 – 0.90) | 1.21 (0.98 – 1.51) |
| $SpO_2 < 95\%$ (n = 33,482) | 17,109/30,137 (56.8) | 860/2,708 (31.8) | 291/637 (45.7) | 0.35 (0.32 – 0.39) | 0.64 (0.53 – 0.78) | 1.81 (1.46 – 2.24) | 0.40 (0.34 – 0.46) | 0.79 (0.64 – 0.98) | 1.99 (1.59 – 2.49) |
| At least 1 respiratory symptom (n = 37,726) | 30,208/34,002 (88.8) | 2,177/3,021 (72.1) | 490/703 (69.7) | 0.32 (0.29 – 0.36) | 0.29 (0.24 – 0.35) | 0.89 (0.72 – 1.11) | 0.44 (0.37 - 0.53) | 0.43 (0.34 - 0.55) | 0.98 (0.78 – 1.23) |
| Diarrhea (n = 30,907) | 6,387/27,758 (23.0) | 335/2,574 (13.0) | 65/575 (11.3) | 0.50 (0.43 – 0.58) | 0.43 (0.31 – 0.58) | 0.85 (0.60 – 1.19) | 0.61 (0.50 – 0.74) | 0.53 (0.38 – 0.74) | 0.88 (0.61 – 1.25) |
| Vomiting (n = 30,342) | 4,148/27,206 (15.2) | 342/2,566 (13.3) | 45/570 (7.9) | 0.86 (0.74 – 0.99) | 0.48 (0.32 – 0.68) | 0.56 (0.37 – 0.82) | 0.75 (0.60 – 0.92) | 0.47 (0.31 – 0.69) | 0.63 (0.42 – 0.94) |
| Abdominal pain (n = 15,775) | 1,347/14,360 (9.4) | 117/1,175 (10.0) | 20/240 (8.3) | 1.07 (0.83 – 1.36) | 0.88 (0.48 – 1.50) | 0.83 (0.43 – 1.47) | 0.94 (0.65 – 1.36) | 0.75 (0.41 – 1.39) | 0.80 (0.43-1.49) |
| Fatigue (n = 16,216) | 4,383/14,780 (29.7) | 263/1,191 (22.1) | 47/245 (19.2) | 0.67 (0.56 – 0.80) | 0.56 (0.38 – 0.82) | 0.84 (0.54 – 1.27) | 0.95 (0.74 – 1.20) | 0.86 (0.57 – 1.29) | 0.90 (0.58 – 1.40) |
| Anosmia (n = 16,202) | 3,216/14,718 (21.9) | 335/1,224 (27.4) | 51/260 (19.6) | 1.35 (1.15 – 1.58) | 0.87 (0.59 – 1.26) | 0.65 (0.43 – 0.96) | 1.79 (1.42 – 2.26) | 1.26 (0.85 – 1.88) | 0.70 (0.46 – 1.07) |
| Ageusia (n = 16,103) | 3,121/14,634 (21.3) | 297/1,212 (24.5) | 45/257 (17.5) | 1.20 (1.01 – 1.41) | 0.79 (0.52 – 1.15) | 0.66 (0.42 – 0.99) | 1.70 (1.36 – 2.13) | 1.17 (0.78 – 1.77) | 0.69 (0.45 – 1.07) |
| SARI (n = 34,118) | 18,089/30,689 (58.9) | 1,308/2,795 (46.8) | 242/634 (38.2) | 0.61 (0.56 – 0.67) | 0.43 (0.35 – 0.52) | 0.70 (0.56 – 0.87) | 0.78 (0.68 – 0.90) | 0.58 (0.47 – 0.72) | 0.74 (0.59 – 0.93) |
| SARI without fever (n = 35,939) | 24,504/32,353 (75.7) | 1,816/2,925 (62.1) | 351/661 (53.1) | 0.52 (0.48 – 0.58) | 0.36 (0.30 – 0.44) | 0.69 (0.56 – 0.85) | 0.75 (0.65 – 0.87) | 0.57 (0.46 – 0.70) | 0.75 (0.61 – 0.94) |
| Hospital-acquired infection (n = 30,722) | 760/27,508 (2.8) | 37/2,617 (1.4) | 34/597 (5.7) | 0.51 (0.33 - 0.74) | 2.13 (1.35 – 3.21) | 4.21 (2.34 – 7.54) | 0.38 (0.22 - 0.65) | 1.28 (0.78 – 2.11) | 3.38 (1.79 - 6.37) |

PSM, Propensity Score Matching; NPrNPu, neither pregnant nor puerperal; OR, Odds Ratio; 98,33%CI, 98,33% confidence interval; SpO2, oxygen saturation in room air; SARI, temperature ≥ 38°C, cough, and onset in 10 days.

rate of respiratory discomfort and $SpO_2 < 95\%$, as well as a higher likelihood of ICU admission, invasive ventilatory support, and death. Therefore, our study data suggest that puerperal women are at a higher risk of severe outcomes than pregnant women and run as much risk as neither pregnant nor puerperal women.

At the beginning of the COVID-19 pandemic, the initial publications addressing infections in pregnant women were case reports, short case series, and systematic reviews. Given the low

**Table 3. Comparison of outcomes in pregnant, puerperal, and neither pregnant nor puerperal reproductive age women before and after propensity score matching – Brazil, February 17, 2020 – January 02, 2021.**

| Outcome | | Neither pregnant nor puerperal | Pregnant | Puerperal | Group comparison | | | Group comparison after PSM (age + ethnicity + schooling + FUB + chronic cardiovascular disease + asthma + diabetes + immunosuppression + obesity + respiratory symptoms) | | |
|---|---|---|---|---|---|---|---|---|---|---|
| | | n (%) | n (%) | n (%) | Pregnant vs. NPrNPu | Puerperal vs. NPrNPu | Puerperal vs. Pregnant | Pregnant vs. NPrNPu | Puerperal vs. NPrNPu | Puerperal vs. Pregnant |
| | | | | | OR (98,33% CI) | OR (98,33% CI) | OR (98,33% CI) | OR (98,33% CI) | OR (98,33% CI) | OR (98,33% CI) |
| ICU admission (n = 32,769) | | 8014/29,368 (27.3) | 574/2,721 (21.1) | 244/680 (35.9) | 0.71 (0.63 – 0.80) | 1.49 (1.23 – 1.81) | 2.09 (1.67 – 2.61) | 0.58 (0.46 – 0.74) | 1.14 (0.87 – 1.51) | 1.97 (1.55 – 2.50) |
| Ventilatory support (n = 31,457) | No[a] | 11,450/28,199 (40.6) | 1626/2,598 (62.6) | 349/660 (52.9) | - | - | - | - | - | - |
| | Yes, invasive | 3,536/28,199 (12.5) | 209/2,598 (8.0) | 133/660 (20.2) | 0.42 (0.35 – 0.50) | 1.23 (0.96 – 1.58) | 2.96 (2.19 – 4.00) | 0.48 (0.31 – 0.74) | 1.29 (0.89 – 1.85) | 2.71 (1.78 – 4.13) |
| | Yes, noninvasive | 13,213/28,199 (46.9) | 763/2,598 (29.4) | 178/660 (27.0) | 0.41 (0.36 – 0.45) | 0.44 (0.35 – 0.55) | 1.09 (0.85 – 1.39) | 0.59 (0.44 – 0.79) | 0.65 (0.49 – 0.87) | 1.10 (0.81 – 1.50) |
| Death (n = 35,700) | | 4,534/32,081 (14.1) | 181/2,904 (6.2) | 114/715 (15.9) | 0.40 (0.33 – 0.49) | 1.15 (0.89 – 1.47) | 2.85 (2.09 – 3.87) | 0.43 (0.33 – 0.57) | 1.09 (0.81 – 1.45) | 2.51 (1.79 – 3.52) |

[a] Reference category.

PSM, Propensity Score Matching; NPrNPu, neither pregnant nor puerperal; OR, Odds Ratio; 98,33%CI, 98,33% confidence interval; FUB, Federative Unit of Brazil; ICU, Intensive Care Unit.

case numbers, the first impression was that pregnant and puerperal women were not at a higher risk for complications and death than the non-obstetric population [3–9]. As epidemiological weeks passed, new studies reported a greater need for invasive ventilation and ICU and an increased number of deaths in the obstetric population [10, 11, 13, 14, 16, 28].

The United States Centers for Disease Control and Prevention published data from January 22 to October 3, 2020, comprising 1,300,938 women with COVID-19. Of these, 23,434 were symptomatic pregnant women. In that study, pregnant women, as opposed to nonpregnant, ran a higher risk of ICU admission, invasive ventilation, extracorporeal membrane oxygenation (ECMO), and death [13]. In our study, the sample included only hospitalized women with SARS-CoV-2. After PSM, our results differed from the CDC results as follows: pregnant women were less likely to be admitted to an ICU (OR 0.58), to need invasive ventilation (OR 0.48), or to die (OR 0.43) than nonpregnant; however, puerperal women were at a higher risk of death than pregnant women. Among all the signs and symptoms reported in the CDC study, cough, headache, myalgia, and fever were the most common and they were mostly reported by the nonpregnant women [13]. Our data also show that the most common signs and symptoms in all groups were cough, dyspnea, and fever, with both pregnant and puerperal women showing fewer symptoms than neither pregnant nor puerperal women. Nevertheless, pregnant women had anosmia and dysgeusia significantly more frequently than neither pregnant nor puerperal women.

In a study with the Brazilian population involving 2,475 pregnant and puerperal women with SARS, 72% of whom had COVID-19 confirmed by RT-PCR, 590 had unfavorable outcomes. The risk increased 2.4 times when the SARS notification occurred in the postpartum period rather than during pregnancy [21]. In the present study, all cases were confirmed by RT-PCR, and the unfavorable outcomes (ICU, mechanical ventilation, and death) were analyzed separately for each group of women (pregnant, puerperal, and neither pregnant nor

**Table 4. Comparison of the death rates of pregnant, puerperal, and neither pregnant nor puerperal reproductive age women before and after propensity score matching according to comorbidities, ICU admission, and ventilatory support – Brazil, February 17, 2020 – January 02, 2021.**

| Variable | Death rate | | | Group comparison | | | Group comparison after PSM (age + ethnicity + schooling + FUB + chronic cardiovascular disease + asthma + diabetes + immunosuppression + obesity + respiratory symptoms) | | |
| --- | --- | --- | --- | --- | --- | --- | --- | --- | --- |
| | Neither pregnant nor puerperal | Pregnant | Puerperal | Pregnant vs. NPrNPu | Puerperal vs. NPrNPu | Puerperal vs. Pregnant | Pregnant vs. NPrNPu | Puerperal vs. NPrNPu | Puerperal vs. Pregnant |
| | n (%) | n (%) | n (%) | OR (98,33% CI) | OR (98,33% CI) | OR (98,33% CI) | OR (98,33% CI) | OR (98,33% CI) | OR (98,33% CI) |
| Chronic cardiovascular disease (n = 5,349) | 1,159/5,093 (22.8) | 24/186 (12.9) | 15/70 (21.4) | 0.51 (0.29 – 0.83) | 0.93 (0.44 - 1.82) | 1.84 (0.74 – 4.39) | 0.47 (0.25 – 0.89) | 0.96 (0.46 – 2.00) | 2.02 (0.82 – 4.99) |
| Asthma (n = 1,804) | 254/1,658 (15.3) | 8/119 (6.7) | 6/27 (22.2) | 0.41 (0.15 – 0.90) | 1.61 (0.44 – 4.57) | 3.93 (0.87 – 16.64) | 0.56 (0.16 – 1.99) | 2.68 (0.80 – 8.94) | 4.79 (0.95 – 24.10) |
| Diabetes (n = 4,539) | 1,143/4,268 (26.8) | 27/212 (12.7) | 19/59 (32.2) | 0.40 (0.24 – 0.64) | 1.30 (0.64 – 2.51) | 3.24 (1.39 – 7.46) | 0.43 (0.23 - 0.81) | 1.52 (0.75 – 3.06) | 3.50 (1.45 – 8.47) |
| Immunodepression (n = 1,213) | 408/1,159 (35.2) | 6/36 (16.7) | 3/18 (16.7) | 0.38 (0.11 – 1.01) | 0.38 (0.06 – 1.48) | 1.02 (0.12 – 6.44) | 0.54 (0.15 – 1.85) | 0.55 (0.12 – 2.61) | 1.03 (0.15 – 7.09) |
| Chronic Renal disease (n = 995) | 385/963 (40.0) | 2/21 (9.5) | 4/11 (36.4) | 0.17 (0.01 – 0.75) | 0.87 (0.15 – 3.93) | 4.96 (0.49 – 87.60) | 0.59 (0.09 – 3.97) | 1.52 (0.30 – 7.54) | 2.56 (0.24 – 26.90) |
| Obesity (n= 3,500) | 762/3,327 (22.9) | 19/125 (15.2) | 15/48 (31.2) | 0.61 (0.32 – 1.07) | 1.54 (0.69 – 3.19) | 2.52 (0.95 – 6.62) | 0.80 (0.37 – 1.76) | 2.29 (1.06 – 4.93) | 2.84 (0.98 – 8.21) |
| Invasive respiratory support (n = 3,878) | 2,249/3,536 (63.6) | 104/209 (49.8) | 77/133 (57.9) | 0.57 (0.40 – 0.80) | 0.79 (0.51 – 1.22) | 1.39 (0.81 -2.38) | 0.68 (0.33 – 1.37) | 0.83 (0.41 – 1.68) | 1.22 (0.67 – 2.22) |
| Noninvasive respiratory support (n = 14,154) | 1246/13,213 (9.4) | 37/763 (4.8) | 18/178 (10.1) | 0.49 (0.32 – 0.72) | 1.09 (0.57 – 1.90) | 2.21 (1.04 – 4.48) | 0.48 (0.27 – 0.86) | 0.92 (0.47 – 1.80) | 1.93 (0.89 – 4.17) |
| ICU admission (n = 8,832) | 2,755/8,014 (34.4) | 127/574 (22.1) | 90/244 (36.9) | 0.54 (0.42 – 0.69) | 1.12 (0.80 – 1.53) | 2.06 (1.38 – 3.06) | 0.69 (0.44 – 1.07) | 1.27 (0.81 – 1.98) | 1.84 (1.18 – 2.86) |

PSM, Propensity Score Matching; NPrNPu, neither pregnant nor puerperal; OR, Odds Ratio; 98,33%CI, 98,33% confidenceinterval; FUB, Federative Unit of Brazil; ICU, Intensive Care Unit.

puerperal) to identify the specific risks for each outcome and each group. After pairing with PSM, puerperal women were more likely than pregnant women to be admitted to ICU (OR 1.97), to receive invasive respiratory support (OR 2.71), to die (OR 2.51), and to acquire the COVID-19 infection in hospital (OR 3.38).

Some hypotheses can explain the worst prognosis for postpartum women. The puerperium is considered a period of high risk for the occurrence of thromboembolism, such as COVID-19, which can have an additive effect on these occurrences [29, 30]. In addition, C-sections increase the risk of maternal mortality and, in Brazil, more than 55% of births occur through this mode [31, 32]. With COVID-19, the risk may be strengthened. Another hypothesis is the three-delays model [33]. During the puerperium, it is common for pregnant women to neglect their own health care because they are focused on taking care of their newborn, reflecting delay in deciding to seek medical assistance, which is compatible with the higher frequency of O2 saturation lower than 95% at admission when compared to pregnant women [34].

In Mexico, a study including a cohort of 5,183 pregnant women and 175,905 nonpregnant women with COVID-19 compared the two groups regarding death, pneumonia, invasive respiratory support, and ICU admission. The data (comorbidities, age, language, and health insurance level) were analyzed with and without adjustment for propensity score matching. After pairing, pregnant women showed a higher likelihood of death (OR 1.84), pneumonia (OR 1.86), and ICU admission (OR 1.86) than nonpregnant, but both groups ran a similar risk

of invasive respiratory support (OR 0.93) [14]. The study, however, did not evaluate either group separately. In contrast to the Mexican study, our data, following adjustments, revealed that pregnant women had a lower likelihood of death (OR 0.43), invasive respiratory support (OR 0.48), and ICU admission (OR 0.58) than both neither pregnant nor puerperal women and puerperal women. It should be mentioned that the two studies are not comparable, given that the Mexican study, on the one hand, included women who were not hospitalized and, on the other hand, divided reproductive age women into only two groups, pregnant and nonpregnant.

Our study has the following strong points: 1) the use of a national database with a large sample size number and no duplicates; 2) patients who were hospitalized due to severe acute respiratory syndrome, which was confirmed by the RT-PCR laboratory test; 3) discrimination between pregnant and puerperal women for a more accurate analysis of the obstetric population; 4) the use of paired comparison analysis through propensity score matching, which allowed adjustments for the demographic data and comorbidities to evaluate symptoms and outcomes.

As limitations, our study included the cases that are notified (hospitalized cases with SARS due to COVID-19), then we could not compare among those infected with COVID-19 in the general population (not hospitalized). Besides, the notification of COVID-19 hospital admissions is compulsory in Brazil, but we cannot guarantee that all patients with COVID-19 who were hospitalized were included and that bias due to missing variables or inaccurately filled fields could not be eliminated.

There is no information on the obstetric characteristics of the pregnant patients, such as gestational age, delivery mode, comorbidities, and perinatal data in the database. Besides, as the data are anonymous, we cannot do the deterministic linkage with the public database of birth registers and mode of delivery.

Furthermore, hospital admission of pregnant women with laboratory-confirmed SARS-CoV-2 infection may be not only because of SARS-CoV-2 symptoms, but also for preventive/cautionary reasons as, in our study, pregnant women had less signs and symptoms at admission.

In the present study, the death risk of each comorbidity was identified separately for the three groups of women: pregnant, puerperal, and neither pregnant nor puerperal. This enabled the use of ORs included in the risk calculations for the COVID-19 progression at hospital admission.

Since puerperal women were at a higher risk for the most severe outcomes (need for ICU, use of invasive ventilatory support, and death), the fact that pregnancy is over must not underestimate the severity risks. Thus, in those cases in which the SARS-CoV-2 infection is acquired at the end of pregnancy, thedelivery should be considered only after overcoming the disease. When it is acquired during the puerperal period, health professionals should remain alert to the severity-related risks. It would be ideal if all women were vaccinated to minimize the risks of SARS-CoV-2 infection and if they adhered to protection measures to prevent contamination by the virus.

As puerperal turned out to be a higher risk group than pregnant women among those hospitalized due to COVID-19, we need more studies comparing these groups. Furthermore, it should be considered that childbirth might influence the progression of COVID-19.

## Author Contributions

**Conceptualization:** Fabiano Elisei Serra, Rossana Pulcineli Vieira Francisco, Agatha Sacramento Rodrigues.

**Data curation:** Fabiano Elisei Serra, Rossana Pulcineli Vieira Francisco, Agatha Sacramento Rodrigues.

**Formal analysis:** Agatha Sacramento Rodrigues.

**Funding acquisition:** Rossana Pulcineli Vieira Francisco, Agatha Sacramento Rodrigues.

**Investigation:** Fabiano Elisei Serra.

**Methodology:** Fabiano Elisei Serra, Rossana Pulcineli Vieira Francisco, Agatha Sacramento Rodrigues.

**Project administration:** Fabiano Elisei Serra.

**Resources:** Fabiano Elisei Serra.

**Software:** Rossana Pulcineli Vieira Francisco, Agatha Sacramento Rodrigues.

**Supervision:** Rossana Pulcineli Vieira Francisco.

**Validation:** Fabiano Elisei Serra, Patricia de Rossi.

**Visualization:** Fabiano Elisei Serra, Rossana Pulcineli Vieira Francisco, Patricia de Rossi, Maria de Lourdes Brizot, Agatha Sacramento Rodrigues.

**Writing – original draft:** Fabiano Elisei Serra, Rossana Pulcineli Vieira Francisco, Patricia de Rossi, Maria de Lourdes Brizot, Agatha Sacramento Rodrigues.

**Writing – review & editing:** Fabiano Elisei Serra, Rossana Pulcineli Vieira Francisco, Patricia de Rossi, Maria de Lourdes Brizot, Agatha Sacramento Rodrigues.

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
