## [Decision Letter · Decision Letter 0]

12 Aug 2021

PONE-D-21-21318

COVID-19 outcomes in hospitalized puerperal, pregnant, and neither pregnant nor puerperal women: a population study

PLOS ONE

Dear Dr. Rodrigues,

Thank you for submitting your manuscript to PLOS ONE. After careful consideration, we feel that it has merit but does not fully meet PLOS ONE’s publication criteria as it currently stands. Therefore, we invite you to submit a revised version of the manuscript that addresses the points raised during the review process.

We look forward to receiving your revised manuscript.

Kind regards,

Sinan Kardeş, M.D.

Academic Editor

PLOS ONE

Journal Requirements:

“This work was supported, in whole or in part, by the Bill & Melinda Gates Foundation [INV-027961]. Under the grant conditions of the Foundation, a Creative Commons Attribution 4.0 Generic License has already been assigned to the Author Accepted Manuscript version that might arise from this submission.”

We note that you have provided funding information within the Acknowledgements Section. Please note that funding information should not appear in the Acknowledgments section or other areas of your manuscript. We will only publish funding information present in the Funding Statement section of the online submission form.

“This work was supported, in whole or in part, by the Bill & Melinda Gates Foundation [INV-027961]. Under the grant conditions of the Foundation, a Creative Commons Attribution 4.0 Generic License has already been assigned to the Author Accepted Manuscript version that might arise from this submission.”

Reviewers' comments:

Reviewer's Responses to Questions

**Comments to the Author**

1. Is the manuscript technically sound, and do the data support the conclusions?

Reviewer #1: Partly

Reviewer #2: Yes

2. Has the statistical analysis been performed appropriately and rigorously? 

Reviewer #1: No

Reviewer #2: Yes

3. Have the authors made all data underlying the findings in their manuscript fully available?

Reviewer #1: Yes

Reviewer #2: Yes

4. Is the manuscript presented in an intelligible fashion and written in standard English?

Reviewer #1: Yes

Reviewer #2: Yes

5. Review Comments to the Author

Reviewer #1: Major issues:

1. Comparing pregnant/puerperal women with other women from 10 to 49 years seems a bit odd. What was the hypothesis behind the study?

2. Details of the PSM should be provided: was it one-to-one matching? Which algorithm was used-Greedy neighbour? The standardized difference after the matching should be reported.

3. Also, wherever multiple comparisons have been done, correction for multiple comparisons should be made.

Abstract:

More numerical data is preferable in the abstract.

What about caesarean section rates? This can be mentioned in the abstract too.

Methods:

The limitation of the Kruskal Wallis is that we cannot find which group is causing the difference. It may be supplemented by comparing groups two by two.

Results:

Table two should mention how many patients were included after matching and also the standardized differences.

“A remarkable fact related to hospital-acquired COVID-19 is that pregnant women developed the disease less frequently (1.4%) than neither pregnant nor puerperal women (2.8%) and puerperal women (5.7%)”. This is possibly due to Berkson's bias.

What was the percentage of missing data and how was it dealt with?

Reviewer #2: Dear Author,

Nice paper, Please write the limitations section in more detail. Especially the limitations related to the patients should be stated very well. similarly, the material method should be written in more detail and the properties examined should be explained. In addition, it should be stated why ethical committee approval was not obtained even though an important data was shared, even if it was a retrospective data.

Best regards

6. PLOS authors have the option to publish the peer review history of their article (what does this mean?). If published, this will include your full peer review and any attached files.

Reviewer #1: No

Reviewer #2: **Yes: **Pinar Yalcin Bahat

---

## [Author Response · Author response to Decision Letter 0]

27 Sep 2021

Responses to the Reviewers:

Reviewer #1: Major issues:

1. Comparing pregnant/puerperal women with other women from 10 to 49 years seems a bit odd. What was the hypothesis behind the study?

Response: Thank you for your question. 

Despite the World Health Organization defines women of reproductive age as “all women aged 15–49 years, we have chosen this comparison because in Brazil, since 1996, women of reproductive age refers to all women aged 10-49 years because in our country there is a significant number of pregnancies in patients under 15 years old. 

This decision is supported by the World Health Organization: "in some estimates from censuses and surveys, the upper age is taken as 44 years and the last age group is thus 40–44 years. More recently, it has been recommended that total fertility rates be shown both by age 15–44 and by age 15–49 years, especially when survey data are used. It is common to add births to girls under 15 years of age to the 15–19-year age group and those to women over 49 years to the 45–49-year age group.” 

References: 

World Health Organization. Reproductive health indicators: guidelines for their generation, interpretation, and analysis for global monitoring. Geneva, WHO press, 2006.

 Secretaria de Saúde da Bahia. Óbitos de MIF e óbitos maternos – Nota Técnica. Available at: http://www3.saude.ba.gov.br/cgi/sim/docs/NT_Obitos_MIF_Mat.pdf (accessed 09/24/2021). 

2. Details of the PSM should be provided: was it one-to-one matching? Which algorithm was used-Greedy neighbour? The standardized difference after the matching should be reported.

Response: Thank you for your comment. You are right. More information about PSM is needed. Details of the PSM are provided in the current version of the manuscript in Data Analysis Section: “Propensity score matching (PSM) was used for estimating and assessing balancing weights for the observations to make the three balanced groups in relation to the confounding variables through Inverse Probability of Treatment Weighting Method (IPTW) [25]. Multinomial regression PS is the method used to create the propensity score weights and the Average Treatment Effect (ATE) is estimated for treatment effects based on IPTW. After estimating the weights, weighted logistic regression is considered for binary variables, and the weighted multinomial log-linear model is fitted for multiple categories variables (as ventilator support). The first analysis, used to compare the symptoms between groups, included age, ethnicity, chronic cardiovascular disease, asthma, diabetes, immunosuppression, and obesity as control variables. The second one, comparing the outcomes, included the same variables plus schooling, Federative Unit of Brazil, and respiratory symptoms. The groups are balanced with respect to the control variables after considering the PSM weight. The results of PSM and standardized difference after the matching can be seen in the supplemental material.The analyses were performed with the statistical R software (R Foundation for Statistical Computing Platform, version 4.0.3) [26] and PSM was carried out with the R Weightlt package [27].”

The standardized difference after the matching is reported in the supplementary material. 

3. Also, wherever multiple comparisons have been done, correction for multiple comparisons should be made.

Response: Thank you very much for your comment. The confidence interval for OR is now considering alpha adjustment for multiple comparisons. The Bonferroni method was considered. As the significance level adopted is 5%, the adjusted alpha is 0.05/3=0.016. All confidence intervals in the tables and the interpretations in the text have also been updated in the current version of the manuscript. 

4. Abstract: More numerical data is preferable in the abstract.

Response: Thank you for your advice. More numerical data were included in the abstract as below (the manuscript is already updated):

“Objective To compare hospitalized reproductive age women with COVID-19 who were pregnant, puerperal, or neither one nor the other in terms of demographic and clinical characteristics and disease progression using Brazilian epidemiological data.

Methods A retrospective analysis of the records of the Information System of the Epidemiological Surveillance of Influenza of the Health Ministry of Brazil was performed. It included the data of female patients aged 10 to 49 years hospitalized because of severe COVID-19 disease (RT-PCR+ for SARS-CoV-2), from February 17, 2020 to January 02, 2021. They were separated into 3 groups: pregnant, puerperal, and neither pregnant nor puerperal. General comparisons and then adjustments for confounding variables (propensity score matching [PSM]) were made, using demographic and clinical characteristics, disease progression (admission to the intensive care unit [ICU] and invasive or noninvasive ventilatory support), and outcome (cure or death). Deaths were analyzed in each group according to comorbidities, invasive or noninvasive ventilatory support, and admission to the ICU.

Results As many as 40,640 reproductive age women hospitalized for COVID-19 were identified: 3,372 were pregnant, 794 were puerperal, and 36,474 were neither pregnant nor puerperal. Groups were significantly different in terms of demographic data and comorbidities (p<0.0001). Pregnant and puerperal women were less likely to be symptomatic than the women who were neither one nor the other (72.1%, 69.7% and 88.8%, respectively). Pregnant women, however, had a higher frequency of anosmia, and ageusia than the others. Puerperal women had a worse prognosis than pregnant women with respect to admission to the ICU, invasive ventilatory support, and death, with OR (95% CI) 1.97 (1.55 – 2.50), 2.71 (1.78 – 4.13), and 2.51 (1.79 – 3.52), respectively.

Conclusion Puerperal women were at a higher risk for serious outcomes (need for the ICU, need for invasive and noninvasive ventilatory support, and death) than pregnant women.” 

5. What about caesarean section rates? This can be mentioned in the abstract too.

Response: Thank you for this comment. We used the data available in SIVEP-Gripe from Brazil; unfortunately, the information about cesarean rates could not be retrieved as they are not available on that database. 

6. Methods: The limitation of the Kruskal Wallis is that we cannot find which group is causing the difference. It may be supplemented by comparing groups two by two.

Response: Thank you very much for your comment. Pairwise comparisons using Wilcoxon rank sum test with continuity correction were considered when the Kruskal Wallis test is used. We added the following text in the current version of the manuscript in Data Analysis Section: “The Nonparametric Kruskal-Wallis test was applied to compare the three study groups in terms of quantitative variables and pairwise comparisons using Wilcoxon rank sum test with continuity correction were considered.”

In Result Section, we have added the multiple comparison for age (only variable we consider the Kruskal-Wallis test): “All three groups are different for age (neither pregnant nor puerperal vs pregnant: p-value <0.0001, neither pregnant nor puerperal vs puerperal: p-value <0.0001 and pregnant versus puerperal: p-value=0.034)”.

7. Results: Table two should mention how many patients were included after matching and the standardized differences.

Response: Thank you for your note. As better explained in Methods Section now, we consider the Inverse Probability of Treatment Weighting Method for estimating and assessing balancing weights for all the observations. In this way, all observations are considered in the analysis after matching but with different weights.

8. “A remarkable fact related to hospital-acquired COVID-19 is that pregnant women developed the disease less frequently (1.4%) than neither pregnant nor puerperal women (2.8%) and puerperal women (5.7%)”. This is possibly due to Berkson's bias.

Response: Thank you for the question. The highest frequency of hospital-acquired Covid-19 disease occurred in the postpartum group of women. A possible explanation for this fact may be the exposure of these women to Covid-19 during hospitalization for delivery, since testing on admission for childbirth has not been universally incorporated in Brazil, making it impossible to identify possible asymptomatic cases. The other groups have a lower risk of exposure to hospital admissions. 

9. What was the percentage of missing data and how was it dealt with?

Response: Thank you for the question. The percentage of missing data depends on the variable in question. In this study, we only worked with valid responses, and we have always informed the number of cases of the variable in question in the tables. For example, for the variable obesity in the pregnant group, the information is 143/1,097 (Table 1). Thus, there are 143 people identified as obese in this group with 1097 valid observations (with information on obesity). This information has been included in Methods Section in the current version of the manuscript. 

Reviewer #2: 

1. Please write the limitations section in more detail. Especially the limitations related to the patients should be stated very well. 

Response: Thank you for your suggestion. We added information about the limitations in the manuscript with the track changes which are also written below:

“As limitations, our study included the cases that are notified (hospitalized cases with SARS due to COVID-19), then we could not compare among those infected with COVID-19 in the general population (not hospitalized). Besides, the notification of COVID-19 hospital admissions is compulsory in Brazil, but we cannot guarantee that all patients with COVID-19 who were hospitalized were included and that bias due to missing variables or inaccurately filled fields could not be eliminated. 

There is no information on the obstetric characteristics of the pregnant patients, such as gestational age, delivery mode, comorbidities, and perinatal data in the database. Besides, as the data are anonymous, we cannot do the deterministic linkage with the public database of birth registers and mode of delivery. 

Furthermore, hospital admission of pregnant women with laboratory-confirmed SARS-CoV-2 infection may be not only because of SARS-CoV-2 symptoms, but also for preventive/cautionary reasons as, in our study, pregnant women had less signs and symptoms at admission.”

2. Similarly, the material method should be written in more detail and the properties examined should be explained. 

Response: Thank you! We have updated the information in “Material and Methods”, also included some suggestions made by the Reviewer #1. The “Results” had to be updated after the suggestions. It can be seen in the revised manuscript with track changes.

3. In addition, it should be stated why ethical committee approval was not obtained even though an important data was shared, even if it was a retrospective data.

Response: Thank you for your note. As it can be seen in the manuscript (also copied below), we explained why it is not necessary in Brazil. 

“SIVEP-Gripe records are publicly available anonymized data. Therefore, according to Brazilian Ethics regulatory requirements, there is no need for ethical approval by an Institutional Review Board.”

The Resolution can be seen below:

“Esta Resolução dispõe sobre as normas aplicáveis a pesquisas em Ciências

Humanas e Sociais cujos procedimentos metodológicos envolvam a utilização de dados

diretamente obtidos com os participantes ou de informações identificáveis ou que

possam acarretar riscos maiores do que os existentes na vida cotidiana, na forma

definida nesta Resolução.

Parágrafo único. Não serão registradas nem avaliadas pelo sistema CEP/CONEP:

(...)

V - pesquisa com bancos de dados, cujas informações são agregadas, sem

possibilidade de identificação individual.”

Translated to English:

"This Resolution provides the standards applicable to research in Human and Social Sciences which methodological procedures involve the use of data directly obtained from the participants or from identifiable information or which may entail greater risks than those existing in everyday life, in the form defined in this Resolution. 

Single paragraph. Will not be registered or evaluated by the Ethical Committee:

(…)

V - search with databases which information is aggregated without possibility of individual identification.”

Reference: Brasil. Ministério da Saúde. Conselho Nacional de Saúde. Resolução nº 510, de 7 de abril de 2016. Diário Oficial da União. Brasília, 24/05/2016. Available at: https://www.in.gov.br/materia/-/asset_publisher/Kujrw0TZC2Mb/content/id/22917581 (accessed 09/24/2021).

---

## [Decision Letter · Decision Letter 1]

8 Oct 2021

PONE-D-21-21318R1COVID-19 outcomes in hospitalized puerperal, pregnant, and neither pregnant nor puerperal women: a population studyPLOS ONE

Dear Dr. Rodrigues,

Thank you for submitting your manuscript to PLOS ONE. After careful consideration, we feel that it has merit but does not fully meet PLOS ONE’s publication criteria as it currently stands. Therefore, we invite you to submit a revised version of the manuscript that addresses the points raised during the review process.

We look forward to receiving your revised manuscript.

Kind regards,

Sinan Kardeş, M.D.

Academic Editor

PLOS ONE

Journal Requirements:

Reviewers' comments:

Reviewer's Responses to Questions

**Comments to the Author**

1. If the authors have adequately addressed your comments raised in a previous round of review and you feel that this manuscript is now acceptable for publication, you may indicate that here to bypass the “Comments to the Author” section, enter your conflict of interest statement in the “Confidential to Editor” section, and submit your "Accept" recommendation.

Reviewer #1: All comments have been addressed

Reviewer #2: All comments have been addressed

2. Is the manuscript technically sound, and do the data support the conclusions?

Reviewer #1: Yes

Reviewer #2: Yes

3. Has the statistical analysis been performed appropriately and rigorously? 

Reviewer #1: Yes

Reviewer #2: Yes

4. Have the authors made all data underlying the findings in their manuscript fully available?

Reviewer #1: Yes

Reviewer #2: Yes

5. Is the manuscript presented in an intelligible fashion and written in standard English?

Reviewer #1: Yes

Reviewer #2: Yes

6. Review Comments to the Author

Reviewer #1: Thank you for the clarifications.

Title: is still difficult to understand. “a population study” provides no additional meaning.

Abstract:

Please separate factor matched for from outcomes.

It is not clear if the outcomes mentioned in the abstract are after propensity matching.

The discussion should provide some hypothesis why puerperal women had poorer outcomes.

Reviewer #2: (No Response)

7. PLOS authors have the option to publish the peer review history of their article (what does this mean?). If published, this will include your full peer review and any attached files.

Reviewer #1: No

Reviewer #2: **Yes: **Pınar Yalçın Bahat

---

## [Author Response · Author response to Decision Letter 1]

24 Oct 2021

Responses to the Reviewers:

Reviewer #1: Minor issues:

1. Title: is still difficult to understand. “a population study” provides no additional meaning.

Response: Thank you for your comment. We agree with you. We decided to exclude “a population study” from the title and keep as “COVID-19 outcomes in hospitalized puerperal, pregnant, and neither pregnant nor puerperal women”.

2. Abstract:

Please separate factor matched for from outcomes.

It is not clear if the outcomes mentioned in the abstract are after propensity matching.

Response: Thank you for your comment. The outcomes results mentioned in the abstract are obtained after propensity matching and this information is presented in the current version of the manuscript. 

3. The discussion should provide some hypothesis why puerperal women had poorer outcomes.

Response: Thank you for this comment as well. We added more hypotheses in the manuscript, also written below:

“Some hypotheses can explain the worst prognosis for postpartum women. The puerperium is considered a period of high risk for the occurrence of thromboembolism, such as COVID-19, which can have an additive effect on these occurrences [29,30]. In addition, C-sections increase the risk of maternal mortality, and, in Brazil, more than 55% of births occur through this mode [31,32]. With COVID-19, the risk may be strengthened. Another hypothesis is the three-delays model [33]. During the puerperium, it is common for pregnant women to neglect their own health care because they are focused on taking care of their newborn, reflecting the delay in deciding to seek medical assistance, which is compatible with the higher frequency of O2 saturation lower than 95% at admission when compared to pregnant women [34].”

---

## [Decision Letter · Decision Letter 2]

29 Oct 2021

COVID-19 outcomes in hospitalized puerperal, pregnant, and neither pregnant nor puerperal women

PONE-D-21-21318R2

Dear Dr. Rodrigues,

We’re pleased to inform you that your manuscript has been judged scientifically suitable for publication and will be formally accepted for publication once it meets all outstanding technical requirements.

Kind regards,

Sinan Kardeş, M.D.

Academic Editor

PLOS ONE

Additional Editor Comments (optional):

Reviewers' comments:

Reviewer's Responses to Questions

**Comments to the Author**

1. If the authors have adequately addressed your comments raised in a previous round of review and you feel that this manuscript is now acceptable for publication, you may indicate that here to bypass the “Comments to the Author” section, enter your conflict of interest statement in the “Confidential to Editor” section, and submit your "Accept" recommendation.

Reviewer #1: All comments have been addressed

2. Is the manuscript technically sound, and do the data support the conclusions?

Reviewer #1: Yes

3. Has the statistical analysis been performed appropriately and rigorously? 

Reviewer #1: Yes

4. Have the authors made all data underlying the findings in their manuscript fully available?

Reviewer #1: Yes

5. Is the manuscript presented in an intelligible fashion and written in standard English?

Reviewer #1: Yes

6. Review Comments to the Author

Reviewer #1: No further comments.

7. PLOS authors have the option to publish the peer review history of their article (what does this mean?). If published, this will include your full peer review and any attached files.

Reviewer #1: No

---

## [Editor Report · Acceptance letter]

5 Nov 2021

PONE-D-21-21318R2 

COVID-19 outcomes in hospitalized puerperal, pregnant, and neither pregnant nor puerperal women 

Dear Dr. Rodrigues:

I'm pleased to inform you that your manuscript has been deemed suitable for publication in PLOS ONE. Congratulations! Your manuscript is now with our production department. 

Kind regards, 

on behalf of

Dr. Sinan Kardeş 

Academic Editor

PLOS ONE